# Beyond the Frontline: A Triple-Line Approach of Thoracic Surgeons in Lung Cancer Management—State of the Art

**DOI:** 10.3390/cancers15164039

**Published:** 2023-08-09

**Authors:** Benjamin Bottet, Nicolas Piton, Jean Selim, Matthieu Sarsam, Florian Guisier, Jean-Marc Baste

**Affiliations:** 1Department of General and Thoracic Surgery, Hospital Center University De Rouen, 1 Rue de Germont, F-76000 Rouen, France; benjamin.bottet@chu-rouen.fr (B.B.); matthieu.sarsam@chu-rouen.fr (M.S.); 2Department of Pathology, UNIROUEN, INSERM U1245, CHU Rouen, Normandy University, F-76000 Rouen, France; nicolas.piton@chu-rouen.fr; 3Department of Anaesthesiology and Critical Care, CHU Rouen, F-76000 Rouen, France; jean.selim@chu-rouen.fr; 4INSERM EnVI UMR 1096, University of Rouen Normandy, F-76000 Rouen, France; 5Department of Pneumology, CHU Rouen, 1 Rue de Germont, F-76000 Rouen, France; florian.guisier@chu-rouen.fr; 6Clinical Investigation Center, Rouen University Hospital, CIC INSERM 1404, 1 Rue de Germont, F-76000 Rouen, France

**Keywords:** lung cancer, lung surgery, minimally invasive surgery, sublobar resection, video-assisted thoracoscopic surgery, robotic-assisted thoracoscopic surgery, enhanced recovery after surgery, immunotherapy, targeted therapy

## Abstract

**Simple Summary:**

Lung cancer is a heterogeneous disease, making it a complex and challenging condition to diagnose and treat effectively. However, recent advances have been made in surgery and perioperative management as well as in the emergence of new therapies (targeted therapy and immunotherapy). These novel treatment approaches have fundamentally altered the course of the disease, offering new hope and improved outcomes for patients. While surgery traditionally played a role mainly in the initial phases of lung cancer, its potential benefits are now being considered at various stages of the disease. The objective of this review is to provide a comprehensive description of the latest surgical approaches in lung cancer. We aim to highlight the importance of integrating these modalities within a patient-centered and personalized treatment pathway.

**Abstract:**

Non-small cell lung cancer (NSCLC) is now described as an extremely heterogeneous disease in its clinical presentation, histology, molecular characteristics, and patient conditions. Over the past 20 years, the management of lung cancer has evolved with positive results. Immune checkpoint inhibitors have revolutionized the treatment landscape for NSCLC in both metastatic and locally advanced stages. The identification of molecular alterations in NSCLC has also allowed the development of targeted therapies, which provide better outcomes than chemotherapy in selected patients. However, patients usually develop acquired resistance to these treatments. On the other hand, thoracic surgery has progressed thanks to minimally invasive procedures, pre-habilitation and enhanced recovery after surgery. Moreover, within thoracic surgery, precision surgery considers the patient and his/her disease in their entirety to offer the best oncologic strategy. Surgeons support patients from pre-operative rehabilitation to surgery and beyond. They are involved in post-treatment follow-up and lung cancer recurrence. When conventional therapies are no longer effective, salvage surgery can be performed on selected patients.

## 1. Introduction: State of the Art of Lung Cancer in 2023

Lung cancer was the second most commonly diagnosed cancer and the leading cause of cancer death in 2020, with 2.2 million new cases and 1.8 million deaths [1]. Lung cancer primarily affects men and is often diagnosed at an advanced stage (75%) [2,3,4], resulting in a low 5-year survival rate of only 10 to 20% [1,2]. However, screening programs and improved follow-up strategies have led to earlier detection and reduced mortality rates. Low-dose computed tomography (CT) is an effective screening method for high-risk individuals, such as heavy smokers, and has demonstrated an ability to detect lung cancer at an earlier stage and to decrease mortality [5,6,7].

Lung cancer is a molecularly heterogeneous disease with various subtypes and clinical presentations, making it a complex disease to diagnose and manage [4] (Figure 1). Non-small cell lung cancer (NSCLC) accounts for approximately 80–85% of newly diagnosed cases of lung cancer annually [8]. Lung adenocarcinoma (LUAD) and lung squamous cell carcinoma (SCC) are the most common subtypes. Histology plays a crucial role in the classification and management of NSCLC. The World Health Organization (WHO) classification enables improvement in patient outcomes by providing greater diagnostic accuracy and better therapeutic strategies through more efficient molecular and biomarker testing [4]. The Tumor, Node, Metastasis (TNM) staging system further guides treatment decisions, allowing clinicians to tailor therapies based on the stage of the disease and overall prognosis [9]. In addition, the WHO classification provides guidelines and recommendations regarding the comprehensive evaluation of molecular markers in lung cancer [4].

Advances in molecular biology have improved our understanding of this heterogeneity, particularly in NSCLC, and have revealed oncogenic drivers that can be targeted with specific therapies. Indeed, lung cancer displays one of the highest rates of targetable genetic alterations [10]. The frequency and prevalence of driver gene aberrations differ among LUAD and lung SCC [11,12,13]. The European Society for Medical Oncology [14] and the WHO classification [4] emphasize the systematic assessment of specific molecular alterations in NSCLC, such as genetic mutations (e.g., EGFR, KRAS), fusions (e.g., ALK, ROS1, RET), and protein overexpression such as programmed death-ligand 1 (PD-L1), among others.

Because samples are often small, it is recommended to spare as much tissue for molecular testing as possible and to use only a limited panel of immunohistochemical markers as well as mucin stains to diagnose and subtype NSCLC [4]. Liquid biopsy includes testing on a variety of cancer biomarkers, such as circulating tumor DNA (ctDNA), microRNA, and circulating tumor cells, which can be collected from non-invasive specimens (plasma, serum, urine, etc.) to determine actionable genomic alterations [15,16]. These challenges have prompted the integration of artificial intelligence (AI) in anatomical pathology [17]. AI technologies, such as machine learning and deep learning algorithms, have shown great potential in revolutionizing anatomical pathology by predicting patient prognosis and treatment response based on image analysis, and contributing to personalized medicine approaches [18].

In selected patients, targeted treatments have replaced the empirical use of cytotoxic therapies and offer more effective and tolerable regimens tailored to specific molecular alterations [14]. Most targetable oncogenic alterations occur in LUAD. The most common genetic alterations in LUAD are EGFR and KRAS-activating mutations, followed by, in frequency, ALK and ROS1 fusions, BRAF mutations, MET exon 14 skipping mutations and MET amplifications, RET gene fusions, and HER2 mutations [14]. Despite the initial success of targeted therapies, our ability to achieve durable remission remains limited by the inevitable development of resistance to targeted therapy. Acquired resistance often arises due to the emergence of secondary mutations [19]. To fight resistance to tyrosine kinase inhibitors, next-generation inhibitors have been developed and have shown efficacy in clinical trials [20,21]. Performing new biopsies in cases of recurrence or relapse of lung cancer is of the highest importance, especially in patients harboring known genetic alterations [22,23,24].

Immunotherapy also plays a significant role in the treatment of lung cancer, similar to melanoma, resulting in major improvements in patient survival [25]. First developed for metastatic or locally advanced NSCLC, immunotherapy is now considered even for neoadjuvant and adjuvant therapy [26,27,28]. Neoadjuvant immunotherapy aims to facilitate early development of memory T cells leading to a strong adaptive anti-tumor response, representing an important advantage over adjuvant therapy [29,30,31]. CheckMate-816, the first phase 3 trial comparing the addition of an anti-PD1 monoclonal antibody (nivolumab) to neoadjuvant platinum-doublet chemotherapy, met its primary endpoint of improved pathologic complete response rates with the addition of nivolumab (24.0% vs. 2.2% for chemotherapy alone). Event-free survival was also improved but overall survival data are still not mature enough [32].

Immunotherapy, known for its ability to induce inflammation and immune-related adverse events, can potentially complicate surgical procedures [33]. Moreover, there were concerns that adding immunotherapy to neoadjuvant chemotherapy could potentially increase the risk of adverse effects. However, the results of the CheckMate 816 trial have shown the opposite. In addition surgery was less cancelled in the chemo-immunotherapy group [32].

These new modalities of lung cancer management have not only reshaped the role of pulmonary surgery but also highlight the need for a multidisciplinary approach and personalized treatment plans tailored to each patient’s specific circumstances and disease characteristics.

The objective of this review is to provide a comprehensive description of the latest surgical approaches in NSCLC. Furthermore, we aim to highlight the importance of integrating these modalities within a patient-centered and personalized treatment pathway.

## 2. Advances in Thoracic Surgery—The Evolving Landscape of First-Line Surgical Approaches

### 2.1. Progress in Minimally Invasive Thoracic Surgical Procedures

Since the end of the 1990s [34,35], the development and spread of new minimally invasive techniques in thoracic surgery, such as video-assisted thoracoscopic surgery (VATS) and robotic-assisted thoracoscopic surgery (RATS), have revolutionized patient management. Thoracotomy was for a long time considered the gold standard, but VATS and RATS have now supplanted it in the management of early-stage NSCLC [3,36,37]. By using “small incisions” and without rib spreading, VATS lung resection has shown better short-term outcomes with lower morbidity and mortality rates, a shorter length of hospital stay, and less pain [38,39,40,41,42]. Similarly, RATS lung resection has shown superiority when compared to open surgery [43,44,45]. Regarding long-term outcomes the superiority of VATS or RATS is still debated [44,45,46,47].

Even if VATS and RATS propose better short-term outcomes, it is essential that they provide equal long-term outcomes regarding overall survival and disease-free survival. No difference was reported between minimally invasive techniques and thoracotomy [44,45,46,47,48,49]. Results are debated concerning operative lymph node staging, and nodal upstaging in open surgery, VATS, or RATS [50,51,52,53,54,55,56]. Thanks to advances in anesthesia [57] and surgery, the mortality rate of lung surgery has decreased over the years [58,59]. Today, the 30-day mortality rate has further decreased to 2% after open lobectomy, 1.3% after minimally invasive lobectomy, and less than 1% after segmentectomy [47,60,61].

### 2.2. Innovation in Perioperative Management

#### 2.2.1. Enhanced Recovery after Surgery (ERAS)

The concept of Enhanced Recovery After Surgery (ERAS) was first defined in 1997 by Professor Henrik Kehlet for colorectal surgery [62]. It was based on six pillars: preoperative information and education, attenuation of stress, pain relief, exercise, enteral nutrition, and growth factors. The ERAS program aims to optimize patient management throughout their surgical journey by implementing specific measures in each phase. First, the preoperative phase plays a crucial role in patient preparation. Education and information strategies are implemented to educate patients about the upcoming surgery, the goals of enhanced recovery, and the steps involved. This enables patients to mentally and physically prepare themselves, which can reduce anxiety and promote active participation in their own recovery [63,64]. Moving on to the intraoperative phase, the ERAS program encourages the use of minimally invasive surgical techniques whenever possible. These techniques can lead to smaller incisions, reduced tissue trauma, and faster recovery [3]. Additionally, optimized pain management strategies, such as the use of regional anesthesia or nerve blocks, are employed to minimize postoperative pain and facilitate early mobilization [57]. The postoperative phase of the ERAS program focuses on early recovery and rehabilitation. Early mobilization, including walking and physical therapy, is initiated as soon as possible to prevent complications and improve overall outcomes. The program also emphasizes early initiation of oral intake, gradually advancing from clear liquids to a normal diet, to expedite the return of bowel function. By integrating these elements into the entire perioperative process, the ERAS program in thoracic surgery aims to reduce surgical stress, minimize complications, shorten hospital stays, and enhance overall patient recovery without increased re-admission rates [65,66].

#### 2.2.2. Prehabilitation

The emergence of preoperative rehabilitation has had a profound impact on the field of thoracic surgery, offering numerous benefits for patients undergoing surgical procedures.

One significant advantage is the observed reduction in postoperative complications. Several studies have demonstrated that preoperative exercise training and physiotherapy can improve patients’ functional capacity, respiratory function, and overall physical fitness [67,68,69]. These improvements contribute to a decreased incidence of postoperative complications, such as pneumonia, atelectasis, and respiratory failure.

The timely management of lung cancer is crucial for optimal patient outcomes. However, the requirement of a five-week preoperative rehabilitation period has posed challenges in meeting the recommended treatment timeline [70]. Fortunately, recent studies have shed light on the possibility of shortening the duration of prehabilitation without compromising its effectiveness. Gravier et al. [71] conducted a randomized trial and demonstrated that a three-week regimen of prehabilitation sessions for individuals with NSCLC yielded similar or even better outcomes compared to the traditional five-week program. These findings suggest that a shorter duration of prehabilitation can be equally effective in preparing patients for surgery. Implementing this modified approach can help avoid unnecessary delays in the recommended treatment timeline, facilitating timely and efficient management of lung cancer patients.

Moreover, preoperative rehabilitation has expanded the pool of patients eligible for surgery. Traditionally, patients with poor preoperative spirometric evaluation or pre-existing comorbidities were deemed unsuitable candidates for surgical intervention [72]. However, these recommendations were based on patients mainly treated by open surgery and without a pre-operative rehabilitation program. As previously mentioned, minimally invasive surgery improves postoperative outcomes. The growing emphasis on preoperative rehabilitation also calls for a reevaluation of the traditional criteria used for patient selection before surgery. Among others, a work by our group (Boujibar et al.) [73] highlights the need to update preoperative assessment protocols, particularly for minimally invasive lung surgery. The inclusion of parameters such as performance at stair-climbing tests [68], incremental shuttle walking tests [74], functional capacity, and overall physical fitness can provide a more comprehensive evaluation of patients’ suitability for surgery. These updated criteria can help identify patients who would benefit from preoperative rehabilitation and enable personalized treatment plans to optimize their surgical outcomes (Figure 2).

### 2.3. The Era of Precision in Thoracic Surgery: Customizing Treatment Approaches

#### 2.3.1. The Role of Multimodal Approaches and Preoperative Planning

Integrating the patient in a multimodal approach through “precision surgery” is of utmost importance in the field of thoracic surgery. Lung surgery is not a binary procedure categorized solely as resectable or non-resectable [75]. It is crucial to move beyond indications based solely on respiratory function and to consider various factors (tumor characteristics: size, appearance, localization; patient comorbidities and age, etc.). Adopting a comprehensive approach allows for a more personalized treatment plan [76,77].

Preoperative planning plays a significant role in determining the surgical approach and the type of resection. Indeed, minimally invasive approaches such as VATS and RATS are becoming standard, rendering the palpation of lesions more difficult, not to mention pure ground-glass opacities, which cannot be felt even in open surgery. In the era of sublobar resection, the use of preoperative tracking techniques is becoming essential in some surgeries [78,79,80]. Several techniques have been described, such as the use of methylene blue [81,82], combined with ^99m^Technetium [83], indocyanine green [84], hook wire [85], electromagnetic navigation bronchoscopy (ENB) [85,86], and intraoperative ultrasound [87].

Resection margins in lung cancer surgery serve as a guide for surgical strategy [88].

Furthermore, 3D reconstructions have become an important part of preoperative planning, particularly for sublobar resection [79,80,89]. The technical aspects of the procedure require a thorough understanding of the complex and highly variable pulmonary anatomy, which could be improved by a 3D model. These reconstructions provide a detailed visualization of the patient’s anatomy and surgical margin [80]. By incorporating advanced imaging and reconstruction techniques, surgeons can better navigate the complex anatomy of the lungs, perform precise, targeted resections, and reduce adverse events [90,91].

In summary, adopting a multimodal approach to thoracic surgery, encompassing precision surgery, is essential. This involves moving beyond simplistic categorizations and considering a range of factors when determining surgical indications. Preoperative planning, lesion localization techniques, and three-dimensional reconstruction all play a critical role in ensuring precise surgical interventions and improving patient outcomes (Figure 2).

#### 2.3.2. Sublobar Resection: Wedge Resection and Segmentectomy

Lobectomy has long been considered the gold standard for the management of lung cancer, providing a complete resection of the affected lobe [36]. We are now in an era where CT-based lung cancer screening has revolutionized the detection of “very early” NSCLC. This refers to tumors that are classified as T1a–bN0 (measuring ≤2 cm and node negative), for which more conservative surgeries such as sublobar resection may be proposed. Indeed, recent guidelines and recommendations have emerged suggesting the potential benefits of sublobar resection. Despite resecting less tissue, segmentectomy must lead to a complete resection of the tumor with safe margins and lymph node dissection, providing accurate staging and preserving long-term outcomes. Segmentectomy is indicated not only for compromised patients [92] but also for specific cases and selected patients [3,36,37], with pure ground-glass opacity <2 cm, adenocarcinoma in situ <2 cm, or minimally invasive or invasive adenocarcinoma <2 cm [93], if expected margins are >1 cm or measuring at least the size of the tumor. In these indications, segmentectomy provided the same short- and long-term outcomes as lobectomy [47,60,94].

Two recent studies have brought about a paradigm shift in the management of lung cancer. The first study, conducted by Saji et al., compared segmentectomy and lobectomy for small-sized peripheral NSCLC. After a median follow-up of 7.3 years, no difference was noted in overall survival, although a lower relapse-free survival was observed after segmentectomy. These findings support the consideration of segmentectomy as an alternative surgical approach for patients with small-sized peripheral NSCLC, as it may offer a less extensive procedure while maintaining comparable outcomes [95]. The second study by Altorki et al. reported the results of a multicenter, non-inferiority trial. Eligible patients with peripheral stage IA NSCLC were randomly assigned intraoperatively to undergo either lobectomy or sublobar resection (wedge resection or segmentectomy). After a median follow-up of 7 years, sublobar resection was not inferior to lobectomy in terms of disease-free survival or overall survival [96].

These two studies provide valuable insights into the management of early-stage lung cancer, expanding the options available for surgical resection. They highlight the potential benefits of sublobar resection by providing a more conservative surgical approach while preserving lung function.

## 3. Second Primary Lung Cancer and Recurrence: Approaching the Second Line

### 3.1. Second Primary Lung Cancer: Impact on Survival and Prognosis

Over the last few decades, therapeutic advances have increased the overall 5-year survival rate of patients with lung cancer. Patients are more frequently followed-up, and recurrences can be detected earlier, often before symptoms occur. Indeed, lung cancer survivors are known to have a high risk of developing a second primary lung cancer. Choi et al. conducted a study that revealed that approximately 8.7% of patients with lung cancer had a second lung cancer. Among these second cancers, around 54.6% were detected within the first 5 years following the initial cancer diagnosis [97]. Moreover, in patients who underwent lung cancer surgery, the estimated risk of developing a second cancer was roughly 1–2% per patient-year after resection [98]. Unfortunately, patients with a second lung cancer had a significant decrease in survival compared to those who remained with a single primary lung cancer (HR = 2.12, 95% CI = 2.06 to 2.17; *p* < 0.001). This decrease in survival was more pronounced in patients with early-stage lung cancer and active smokers than in those with advanced cancer and former or non-smokers [97]. It is thus crucial to focus on CT screening and smoking cessation.

Initially documented in 2006 [99], the phenomenon of NSCLC transforming into small-cell lung cancer (SCLC) has now been firmly established. The occurrence of histologic changes in lung cancer following initial diagnosis was attributed to either transformation between NSCLC and SCLC or undetected mixed histology at diagnosis due to tumor heterogeneity [100]. Approximately 10% to 28% of SCLC cases exhibit an NSCLC component [101,102,103]. Distinguishing between transformation and de novo mixed lung cancer histology can pose challenges. To date, EGFR-mutant NSCLC is the most common source of SCLC transformation, significantly higher than ALK-rearranged NSCLC [100,103]. More recent estimates of the frequency of this type of transformation range from 3% to 10% [104]. SCLC transformation is associated with poor prognosis. The estimated median survival was approximately 6 months, which was less than in primary SCLC with extensive disease [105].

### 3.2. Advances in Diagnostic Techniques and Surgical Approaches for Managing Second Lung Cancer

Performing new biopsies has become essential for the management of recurrent and second-line primary lung cancer. Less invasive procedures should be preferred [9,106].

One commonly used technique is bronchoscopic biopsy. Using this minimally invasive procedure, samples can be obtained from the tumor or adjacent areas using specialized tools such as forceps, brushes, or needles. Bronchoscopy may be coupled with a radial ultrasound probe, also known as radial endobronchial ultrasound (r-EBUS). Virtual bronchoscopy software allows one to locate the tumor and identify the optimal bronchial path to the tumor [107]. Endobronchial ultrasound-guided transbronchial needle aspiration or endoscopic ultrasound-guided fine-needle aspiration are used for the diagnosis of mediastinal lymph node metastases [108,109].

Electromagnetic navigation bronchoscopy (ENB) is a minimally invasive procedure used for the diagnosis and staging of lung lesions. During an ENB procedure, a preoperative CT scan is used to create a three-dimensional virtual map of the patient’s lungs. This map serves as a guide for the bronchoscopist to navigate through the airways to the nodule [86]. However, ENB and r-EBUS have some limitations when it comes to pure ground-glass opacities and nodules without bronchial signs [110,111].

When nodules are inaccessible to EBUS, samples can be obtained using a CT scan. CT-guided transthoracic lung biopsy is a minimally invasive diagnostic procedure for tissue diagnosis of peripheral lung nodules [112] and, in some cases, mediastinal metastases [113].

Among surgical techniques, video mediastinoscopy is a safe and effective procedure to evaluate mediastinal lymph nodes surrounding the trachea [114]. Although re-mediastinoscopy can be safely performed in expert centers, less invasive techniques should be preferred in cases of mediastinal recurrence [106].

Although VATS can reach almost all mediastinal lymph node stations, it is the most invasive procedure. However, VATS allows surgeons to access and examine ipsilateral lymph nodes, providing accurate staging information. Additionally, VATS enables meticulous assessment of the pleura, aiding in the detection of metastatic spread. In addition, it is also possible to perform sublobar resection in cases of lung cancer recurrence, providing both sufficient tissue for genetic testing and safe resection margins for curative management. Some authors perform these procedures on an outpatient basis [115,116]. Abid et al. showed that a second surgical resection for a second NSCLC did not result in significantly higher morbidity than the first surgery [117]. Anatomical sublobar resection emerges as a favorable approach, striking a balance between surgical efficacy and preservation of lung function. This approach can also be considered during the first surgery if a suspicious synchronous lesion is identified, which may potentially require surgical intervention at a later stage.

### 3.3. Management of Recurrence after Lung Cancer Treatment

Complete resection remains the most effective treatment for early-stage NSCLC [3,37]. However, despite successful surgeries, recurrence rates of approximately 20% to 50% [118,119,120] pose significant challenges to long-term survival [3,9]. Multiple factors, including TNM stage, surgical approach, resection quality, genetic mutations, and treatment response, influence the likelihood of recurrence [9,14,37,47,95,96]. Accurate diagnosis of recurrence is crucial as it profoundly impacts therapeutic decisions. Sonoda et al. revealed that patients with 1 to 2 recurrences had better survival outcomes than those with more than 3 recurrences, emphasizing the importance of managing recurrent cases carefully [121]. In light of these findings, it is imperative to prioritize localized treatments for patients presenting with 1 to 2 recurrences. Minimally invasive surgical procedures, such as VATS or RATS, offer a good option to treat recurrent lesions [122]. Sublobar resection, including wedge and segmentectomy, is preferred to spare healthy lung tissue. Additionally, non-surgical options such as stereotactic radiotherapy [123] and radiofrequency ablation [124] provide efficient and less invasive alternatives.

## 4. Salvage Surgery in Advanced NSCLC—Third Line

### 4.1. Improved Outcomes in Metastatic Cancers Treated with Immunotherapy

Today, immune checkpoint inhibitor (ICI) treatment, alone or in combination with chemotherapy, is the standard of care for most patients with unresectable NSCLC without targetable molecular alterations [125]. Since their introduction, ICIs have improved the prognosis of advanced NSCLC [126,127,128]. Given the existence of long-term survivors among patients with stage IV NSCLC treated with ICI, lung surgery may be considered in selected cases to improve outcomes in three different scenarios: (i) synchronous oligometastatic disease, (ii) oligopersistence of lung tumor after partial response, (iii) oligoprogression of disease. In some cases, the persistence or growth of pulmonary abnormalities might be due to macroscopic residual disease, granulomatosis reaction, or parenchymal fibrosis with no residual tumor [33]. Anatomical lung resection may be suggested to obtain a pathologic analysis of persisting lung abnormalities and to eradicate any macroscopic residual disease.

### 4.2. Salvage Surgery: Safety and Feasibility

Salvage surgery is defined as lung resection in patients with unresectable or initially metastatic lung cancer who have received previous treatments such as chemotherapy, radiotherapy, targeted therapy, and immunotherapy. Surgical resection is performed more than 12 weeks after the last treatment session, and it is not considered neoadjuvant therapy. When patients received previous high-dose radiotherapy, the delay between lung resection and radiotherapy increased the risk of complications [129]. Immunotherapy generates peritumoral inflammation that may increase tissue adherence, with a potentially higher risk of perioperative complications. Several series showed a higher rate of tissue fibrosis/inflammation [33,130,131,132,133]. Several studies evaluated salvage surgery after immunotherapy in patients with metastatic cancer, demonstrating its safety and feasibility [130,131,132,133,134]. There is much heterogeneity in the results of these studies. Minimally invasive surgery was performed in less than 50% of cases (36–48%) [130,131,134] and in 100% of cases in one study [132]. Postoperative complication rates ranged from 16% to 43% [130,131,132,133,134], with a 90-day mortality rate between 0% and 9% [130,131,132,134]. Nearly 100% of cases achieved complete resection of the tumor [130,131,132,134], while the complete pathologic response rate ranged from 27% to 37.5% [130,132,133,134].

However, these studies had limitations due to their retrospective nature, case series, heterogeneity in surgical procedures and tumor invasion areas. Moreover, the reported cases were often from highly selected patient populations.

### 4.3. Patient Selection Criteria for Salvage or Rescue Surgery

The selection of patients eligible for salvage surgery following immunotherapy +/− chemo- or chemo-radiotherapy treatment requires careful consideration of various factors. Baseline evaluation and staging play a crucial role in determining the suitability of patients for salvage surgery. A comprehensive assessment is typically conducted, which includes an enhanced chest CT scan, brain magnetic resonance imaging (MRI), and abdominal CT as routine imaging modalities. Patients with signs of disease progression or distant metastasis should be excluded, as should those with tumors invading vital structures such as great vessels, diaphragm, heart, trachea, and carina, with a risk of incomplete resection and perioperative complications. Histologic examination and driver mutation analysis are performed through bronchoscopy or subcutaneous needle biopsy. These include confirmed lymph node down-staging assessed by chest CT scan or positron emission tomography (PET)/CT [135]. In cases where there is a bulky mediastinal mass or a need for pathologic confirmation of the N stage, PET/CT and invasive mediastinal staging or EBUS should be used. Resectability of the tumor is then reassessed by a multidisciplinary tumor board composed of thoracic surgeons, oncologists, and radiologists with expertise in the field [75] (Figure 3).

## 5. Conclusions

Overall, the insights gained from these studies and advances in thoracic surgery highlight the importance of early detection, accurate staging, personalized treatment strategies, and multidisciplinary care to optimize outcomes for patients with NSCLC. As we continue to uncover novel therapeutic strategies and refine surgical approaches, a comprehensive and integrated approach involving collaboration among clinicians, surgeons, and researchers will further improve outcomes and the quality of life for patients with lung cancer.

## Figures and Tables

**Figure 1 cancers-15-04039-f001:**
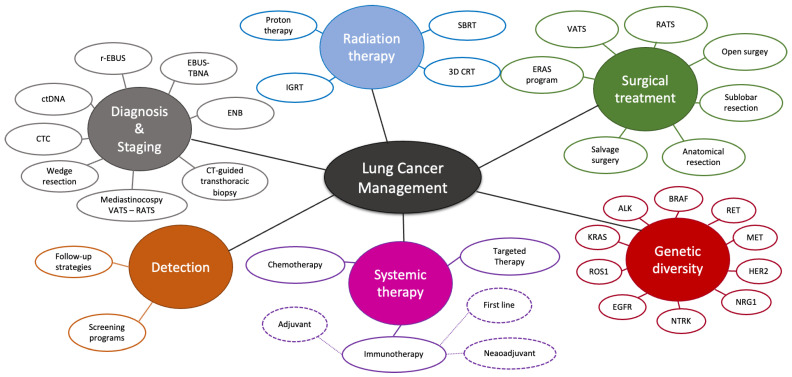
Lung cancer is a heterogeneous disease with a complex and multimodal management. Lung cancer is a highly heterogeneous disease including numerous subtypes. Recent advances in techniques have significantly expanded the treatment landscape, resulting in both increased complexity and the potential for personalized medicine, offering patients the prospect of more effective and precise interventions based on their unique tumor profiles. VATS: video-assisted thoracoscopic surgery; RATS: robotic-assisted thoracoscopic surgery; r-EBUS: radial endobronchial ultrasound; EBUS-TBNA: endobronchial ultrasound-guided transbronchial needle aspiration; ENB: Electromagnetic navigation bronchoscopy; CTC: Circulating tumor cells; IGRT: Image guided radiation therapy; SBRT: Stereotactic body radiation therapy; 3D CRT: 3-Dimensional conformal radiation therapy; ERAS: enhanced recovery after surgery.

**Figure 2 cancers-15-04039-f002:**
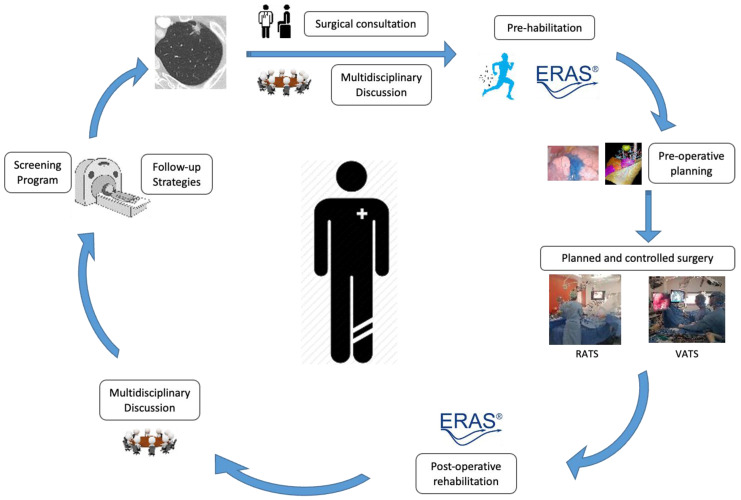
Patient Pathway for Personalized Surgical Management of Early-Stage Lung Cancer in 2023. The patient, initially screened or monitored for a previous cancer, is diagnosed with early-stage lung cancer based on a recent CT scan. In order to determine the best course of treatment, the patient’s case is presented at a multidisciplinary meeting, where a team of specialists collectively discusses and develops a personalized treatment plan. The patient has a consultation with a thoracic surgeon, who explains the personalized surgical approach for their case (3D reconstruction, sublobar resection, and preoperative rehabilitation). The surgery is performed using minimally invasive techniques. The patient is included in an ERAS program. Based on the pTNM staging system, a decision is made regarding the need for further surveillance imaging.

**Figure 3 cancers-15-04039-f003:**
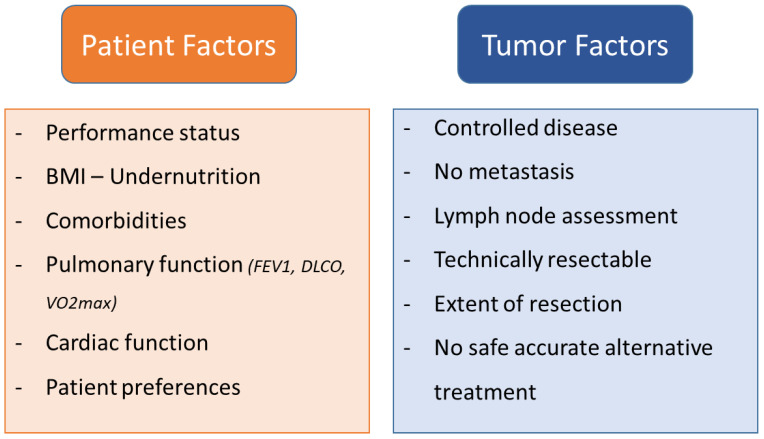
Comprehensive overview of factors to consider and questions to address when considering salvage surgery.

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
