# Peer review of "Beyond the Frontline: A Triple-Line Approach of Thoracic Surgeons in Lung Cancer Management—State of the Art"

_cancers, 2023, doi:10.3390/cancers15164039_

Round 1

Reviewer 1 Report

This review article is a narrative review of “Approach of Thoracic Surgeons in Lung Cancer Management”.

The article is well organized and provides lessons and learned. However, there are a few criticisms.

1.    Figure 1; Circulating tumor cells (CTC) could be added to Diagnosis and staging.

2.    Line 243; What dose "guide+++++++for" mean?

3.    Line 284; CALGB 140503 accumulated "a peripheral lung nodule with a solid component measuring 2 cm or less", thus T1aN0 should be revised.

Author Response

Comment 1: Figure 1; Circulating tumor cells (CTC) could be added to Diagnosis and staging.

Response 1: As requested, CTC has been added in Figure 1.

Comment 2: Line 243; What dose "guide+++++++for" mean?

Response 2:  Sorry, it is a typing error. This problem has been solved.

Comment 3: Line 284; CALGB 140503 accumulated "a peripheral lung nodule with a solid component measuring 2 cm or less", thus T1aN0 should be revised.

Response 3: We agree and we corrected the sentence by changing T1a to Stage IA.

Reviewer 2 Report

Dear Editor and Authors,

Thank you for asking me to review this very interesting review titled “Beyond the Frontline: A Triple-Line Approach of Thoracic Surgeons in Lung Cancer Management – State of the art.” by Dr. Bottet and colleagues from the Department of General and Thoracic Surgery at the CHU Rouen in Rouen, France.

In this very thorough review article the authors discuss modern aspects of thoracic surgery surgical and oncological management of lung cancer such as molecular testing and therapies, immunotherapy, minimally invasive thoracic surgery such as VATS and RATS, enhanced recovery post surgery (ERAS), prehabilitation, EBUS, EUS, sublobar resections, salvage surgery and more.

This is an extensive and thorough review as I mentioned touching in almost, nay all, aspects of modern thoracic surgery and it was a pleasure from my part as a young thoracic surgeon to read it. It is also well written and nicely illustrated which was an extra benefit.

I have very minor corrections to suggest which if corrected I then have no objection to recommend the publication of this work!

Specifically:

1.       In line 243 it seems the authors have accidentally pressed the + button a number of times and this needs to be correcte.

2.       In the same section 2.3.1, the authors do not mention radiolabeling with technicium as a technique to localize and mark small nodules and ground glass opacities. Similarly, intraoperative ultrasound is not mentioned! I suggest they add these two techniques and discuss them, adding some additional appropriate references.

3.       The title of section 3.1 is wrong, what the authors discuss is not recurrence but a second primary. I suggest they discuss the management of recurrence in this section via VATS wedge resection, sublobar resection or RF ablation and add a subsection for the second primary lung cancers! Of course appropriate literature must be cited.

In conclusion, as mentioned this is an excellent work and I support its publication fully provided some minor corrections are made. Good job and kind regards to all.

Language is fine. Minor editing while proofreading is needed.

Author Response

Comment 1: In line 243 it seems the authors have accidentally pressed the + button a number of times and this needs to be correcte.

Response 1:  Sorry, it is a typing error. This problem has been solved.

Comment 2: In the same section 2.3.1, the authors do not mention radiolabeling with technicium as a technique to localize and mark small nodules and ground glass opacities. Similarly, intraoperative ultrasound is not mentioned! I suggest they add these two techniques and discuss them, adding some additional appropriate references.

Response 2:  We changed and updated our references according to your comments.

Comment 3: The title of section 3.1 is wrong, what the authors discuss is not recurrence but a second primary. I suggest they discuss the management of recurrence in this section via VATS wedge resection, sublobar resection or RF ablation and add a subsection for the second primary lung cancers! Of course appropriate literature must be cited.

Response 3: As requested, we changed our titles of section 3. We added a new section dealing with management of recurrence and new references have been cited.